# Stable isotope evidence of anthropocene disruption in African softshell turtle foraging

**Willemien de Kock**[1,2☯*], **Marcel T. J. van der Meer**[3], **Ronald van Bommel**[3],
**Alberto J. Taurozzi**[4], **Matthew Von Tersch**[5], **Morten E. Allentoft**[6,7],
**Meaghan Mackie**[4,8,9,10], **Max Ramsøe**[4], **Matthew Collins**[4,11], **Michelle Alexander**[5],
**Per J. Palsbøll**[2,12], **Canan Çakırlar**[1☯], **Oguz Turkozan**[13☯]

1 Groningen Institute of Archaeology, Faculty of Arts, University of Groningen, Groningen, Netherlands,
2 Marine Evolution and Conservation Group, Groningen Institute for Evolutionary Life Sciences, Faculty
of Science and Engineering, University of Groningen, Groningen, Netherlands, 3 Department of Marine
Microbiology & Biogeochemistry, NIOZ Royal Netherlands Institute for Sea Research, Den Hoorn, The
Netherlands, 4 The Globe Institute, Faculty of Health and Medical Science, University of Copenhagen,
Copenhagen, Denmark, 5 Department of Archaeology, BioArCh, University of York, York, United Kingdom,
6 Trace and Environmental DNA Lab, School of Molecular and Life Sciences, Curtin University, Perth,
Western Australia, Australia, 7 Lundbeck Foundation GeoGenetics Centre, GLOBE Institute, University of
Copenhagen, Copenhagen, Denmark, 8 Novo Nordisk Foundation Center for Protein Research, Faculty of
Health and Medical Science, University of Copenhagen, Copenhagen, Denmark, 9 School of Archaeology,
University College Dublin, Dublin, Ireland, 10 Archaeobiomics, Department of Life Sciences and Systems
Biology, University of Turin, Turin, Italy, 11 Department of Archaeology, University of Cambridge,
Cambridge, United Kingdom, 12 Center for Coastal Studies, Provincetown, Massachusetts, United States
of America, 13 Department of Biology, Adnan Menderes University, Faculty of Science and Arts, Aydın,
Turkey

☯ These authors also contributed to study design.
* willemien@palaeome.org

Marine and Environmental Sciences Centre,
PORTUGAL

**Peer Review History:** PLOS recognizes the
benefits of transparency in the peer review
process; therefore, we enable the publication
of all of the content of peer review and
author responses alongside final, published
articles. The editorial history of this article is
available here: https://doi.org/10.1371/journal.
pone.0339589

## Abstract

We examined the dietary habits of contemporary and Middle to Late Holocene
(ancient) populations of African softshell turtles (*Trionyx triunguis*) from the northern
Levant using stable isotope analysis ($\delta^{13}C$, $\delta^{15}N$, and $\delta^{34}S$) and ZooMS biomarker
identification. Our study presents the first application of ZooMS to this taxon, facili-
tating species-level identification. Stable isotope values point to potential variation in
*T. triunguis* diets, possibly reflecting changing ecosystem conditions. Modern turtles
from the south-western Turkish coast exhibit relatively high $\delta^{15}N$ values, but low $\delta^{13}C$
values, likely influenced by human-provided carrion and agriculture-driven inputs.
Ancient turtles (n = 4) from the Levant exhibit more diverse diets, with two individuals
indicating a more pronounced marine foraging signature. These preliminary find-
ings are consistent with increased anthropogenic influence on *T. triunguis* foraging
in some regions. This study provides new biomolecular insights into the ecological
history of *T. triunguis*, increasing our understanding of its (long-term) dietary plasticity
and potential response to anthropogenic pressures.

**Data availability statement:** The mass spectrometry proteomics data have been deposited to the ProteomeXchange Consortium via the PRIDE partner repository (Perez-Riverol et al. 2019) with the dataset identifier PXD062259.

**Funding:** The first author received support from the Marie Skłodowska-Curie Innovative Training Network SeaChanges, funded by the European Union's Horizon 2020 research and innovation programme (Marie Skłodowska-Curie grant agreement No. 813383), and from the Netherlands Sectorplan Social Sciences and Humanities (SSH, 2022). This study was funded by the Royal Netherlands Academy of Arts and Sciences (KNAW) Ecology Fund under grant number KNAWWF/747/ECO2021-17. There was no additional external funding received for this study.

**Competing interests:** The authors have declared that no competing interests exist.

## Introduction

Turtles are among the most threatened vertebrates groups due to habitat destruction, overexploitation for pets and food, and climate change ([1]; [2]). Turtles play vital ecological roles, including bioturbation of soils, infaunal mining of seabeds, seed dispersal and germination facilitation, nutrient recycling, and consumption [3]. Furthermore, they contribute to regulating water quality in freshwater ecosystems [4]. The African softshell turtle (*Trionyx triunguis*), is a freshwater species found in the Middle East and Africa, particularly in the Levant and the Nile River basin (Fig 1.). It is a member of one of the most threatened turtle clades, and yet one of the least studied turtle species. Although the global population of *T. triunguis* is classified as "Vulnerable" by the IUCN [5] and is protected under the Bern Convention (*Convention on the Conservation of European Wildlife and Natural Habitats*) and CITES (*Convention on International Trade in Endangered Species of Wild Fauna and Flora*), the African population is considered "rare", with some suggesting it warrants a higher IUCN Red List category [6]. In Turkey, major contemporary threats include accidental and intentional fishing, predation, and pollution [7].

Softshell turtles occasionally venture into marine environments [8], raising questions about their reliance on marine resources. Understanding how these turtles' diets have shifted over time, particularly in response to anthropogenic pressures, is crucial for evaluating the current health of vulnerable populations. By studying dietary shifts, we can place their present ecological status within a broader temporal context, illustrating the long-term effects of human activity. The dietary habits of modern African *T. triunguis* populations have been examined in both oil-polluted and pristine habitats, revealing considerable differences in their dietary preferences [9]. However, there is limited evidence regarding the broader food ecology of contemporary *T. triunguis*, which exhibited a more carnivorous diet compared to sympatric pelomedusidae [6,9]. To contextualise these ecological patterns, it is essential to consider the species' life history and its ability to exploit both freshwater and marine environments. Despite their primary freshwater habitat, African Softshell Turtles have demonstrated the ability to exploit marine environments, first documented in Iskenderun Bay in the early 20th century [10]. Their presence as bycatch in the eastern Mediterranean Sea [11,12] suggests that some individuals opportunistically forage in marine settings. Whether such marine foraging behaviour can be detected across time or populations, however, remains uncertain.

There have been no studies addressing whether modern and ancient populations of *T. triunguis* show similar patterns of environmental use or dietary signatures. Any comparison of past and present remains therefore provides an opportunity to investigate potential continuity or divergence in foraging strategies. Moreover, it is not yet clear whether the observed ecological plasticity of *T. triunguis* is a universal trait of the species or one that emerges only in certain local populations. Addressing these uncertainties can provide insight into the functional roles of *T. triunguis* within their ecosystems [3], which may inform conservation and management strategies of this understudied species.

The oldest published Holocene specimen of *T. triunguis* comes from the southern Aegean island of Symi, dating to the 1950s [13]. Archaeological records also indicate their presence in the Levant during the Middle Holocene [14,15], where they were

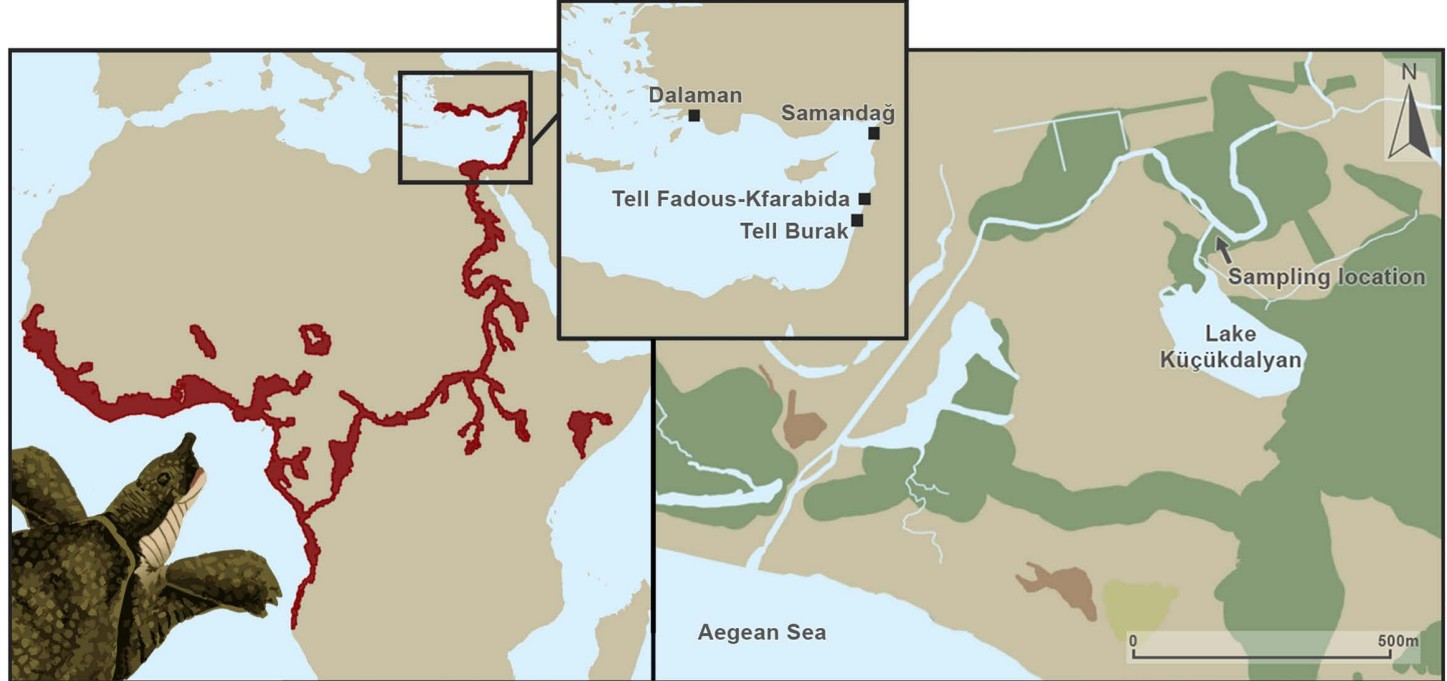

**Fig 1. (Left) The distribution of T. triunguis (red) from [5].** (Centre) Eastern-Mediterranean locations of two modern sites (Dalaman and Samandağ) and two archaeological sites (Tell Fadous-Kfarabida and Tell el-Burak) with *T. triunguis* specimens discussed in this study. (Right) Aerial photograph of the sampling location near Dalaman, Muğla, Turkey. Figure created with assistance from S.E. Boersma.

likely hunted for consumption. However, no archaeological evidence of this species has been found in Cyprus or along the Aegean coast of modern-day Turkey. This absence could suggest that populations in regions such as Dalaman or Samandağ may have become established relatively recently. If *T. triunguis* had been present in antiquity, we would expect to find remains, especially the highly distinguishable shells, or artistic depictions, as seen with sea turtles, which are commonly represented in archaeological contexts [14]. Evidence of human-softshell interactions in the wider region during the Holocene is diverse. In ancient Egypt, *T. triunguis* appears in both literature and art, and was part of the diet of communities along the Nile as far back as the Predynastic (~8–5 kya) period [16]. A striking example of human-softshell turtle interactions comes from the Assyrian site of Kavuşan Höyük (~2.7 kya) located in modern-day eastern Turkey, where *Rafetus euphraticus* (Euphrates softshell turtles) were found interred alongside human burials [17]. These records highlight the potential biogeographic plasticity and the presence of soft shelled turtles in the anthropogenic niche since the middle Holocene, and provide context for studying their past ecological roles and dietary shifts.

Stable isotope analysis provides a powerful means of investigating diet and habitat use in both modern and archaeological specimens. The ratios of stable isotopes (used in this study) expressed as $\delta^{13}C$, $\delta^{15}N$, and $\delta^{34}S$ can reflect different aspects of diet and environmental interaction. $\delta^{13}C$ values distinguish between sources of primary production, such as $C_3$ and $C_4$ plants or marine and freshwater resources, offering insights into habitat and carbon source variability [18; 19]. $\delta^{15}N$ values increase with trophic level, providing information on an organism's position within the food web [20,21]. $\delta^{34}S$ values, are especially useful in aquatic systems because marine environments typically exhibit higher and more homogeneous sulphur isotope values than freshwater or terrestrial ecosystems, thus serving as a sensitive indicator of habitat use [22,15]. When used together, stable isotope measurements can enable broad reconstructions of dietary and ecological variation across space and time.

In this study, we explore these questions through a small but informative dataset (ancient n = 4, modern n = 11), aiming to identify whether stable isotope values from both modern and archaeological *T. triunguis* specimens show evidence of marine foraging and to what extent such signatures are consistent between time periods and regions. To address questions about the dietary habits, ecological interactions, and potential anthropogenic impacts on *T. triunguis*, we combined biomolecular and isotopic approaches. First, we employed ZooMS (Zooarchaeology by Mass Spectrometry, [23]) to identify *T. triunguis* remains from collagen type 1 peptide sequences, as osteological species identification was unfeasible due to high fragmentation. Next, **we compared stable isotope values** ($\delta^{13}$C, $\delta^{15}$N, and $\delta^{34}$S) from newly identified prehistoric bones and freshly sampled modern individuals to assess broad-scale patterns rather than definitive temporal trends. This study represents the first stable isotope and historical ecology analysis of *T. triunguis*, offering preliminary insights into its past and present ecological roles and testing the potential of isotopic data to inform future studies on ecological plasticity in the African softshell turtle.

## Methods

### Sample description

Eleven modern and four ancient *T. triunguis* were sampled in this study. Sampling of eight modern turtles was conducted from August 16th to 20th, 2021, near Dalaman, Muğla, Turkey (Fig 1). Turtles moving between Lake Küçükdalyan and the Aegean Sea pass through this area. Turtles were lured with bait (chicken liver) into a trap formed by three eel fykes arranged in a triangle. Once captured, each turtle was weighed, its curved carapace length and width measured, and its sex recorded, as both size and sex can influence competitive dynamics in feeding, resource access, and overall ecological interactions. A skin tissue sample was taken from the rear end of the carapace using a sterile 0.3 cm punch, immediately stored in ethanol, and exported for stable isotope analysis. Seven turtles (Da2–Da8) were sampled and released back into the water, although turtle Da6 was not weighed. One skin biopsy (Da1) was collected from a recently deceased turtle found near the sampling site, likely bycatch. Additionally, three skin biopsies (Samandağ_1–3) from stranded (deceased) modern *T. triunguis* individuals in Samandağ, Hatay, Turkey, were provided. This study was conducted in accordance with the regulations of the Directorate of Nature Conservation and National Parks of the Ministry of Agriculture and Forestry, Turkey, which granted research permission for the fieldwork. Ethical approval for sampling procedures was obtained from the Animal Experiments Ethics Committee of Aydın Adnan Menderes University (Permit Number: 64583101/2021/039). The biopsy location on each turtle was selected to minimize disturbance, and all handling was performed using clean gloves. Turtles were processed as quickly and gently as possible to reduce stress and were released immediately into their natural environment after sampling, within 5 minutes. No turtles were sacrificed in this study. All eleven skin biopsies were exported to the University of Groningen under CITES permit 22NL305802/11.

We examined one ancient *T. triunguis* bone fragment from the Early Bronze Age settlement Tell Fadous-Kfarabida, situated in the central Levant (Lebanon). Radiocarbon dating and stratigraphy [24] place this specimen to ca. 5–4.5 kya. The other three ancient samples came from Tell el-Burak, ca. 100 km south of Fadous. These remains exclusively belong to the Iron Age II phases A, B, and C, dating to ca. 2.6–2.4 kya [25]. Prior to destructive sampling, all bones were scanned using a three-dimensional scanner and reconstructed to have a record of the unsampled specimen.

### Site geomorphology

The modern sampling site near Dalaman (Muğla, Turkey) is a brackish, riverine environment where turtles move between Lake Küçükdalyan and the Aegean Sea. Samandağ (Hatay, Turkey), the source of additional modern samples, lies at the mouth of the Orontes River, an active freshwater outflow, forming a coastal delta. The archaeological site Tell Fadous-Kfarabida (Lebanon), the Early Bronze Age site, is coastal but lacks evidence for a perennial freshwater source. In contrast, the Iron Age site of Tell el-Burak, located further south, shows greater access to freshwater environments,

as supported by the presence of other freshwater species in the faunal assemblage [26]. Expanded descriptions of the archaeological sites are present in the supplementary information.

### Sample analyses

#### i) *ZooMS*

Species identification with ZooMS analysis of ancient bones was conducted following the protocol of [23] at the Globe Institute, University of Copenhagen. The matrix assisted laser desorption ionisation-time of flight mass spectrometry (MALDI-TOF) measurements were undertaken at the University of Cambridge (Supporting Information S1 File for further details). Comprehensive shotgun proteomic data were also acquired from a museum reference specimen for peptide marker identification, using nanoflow liquid chromatography tandem mass spectrometry (LC-MS/MS) on an EASY-nLC 1200 system coupled to an Exploris 480 mass spectrometer (Thermo Scientific). Detailed mass spectrometry parameters are described in [26]. The mass spectrometry proteomics data have been deposited to the ProteomeXchange Consortium via the PRIDE partner repository [27] with the dataset identifier PXD062259.

#### ii) *Stable isotope analyses*

Collagen extraction from ancient bone samples followed a modified [28,28] and were conducted at BioArCh, University of York, while modern turtle tissue samples were prepared using a standard methodology for sea turtle biopsies [29]. Details are provided in Appendix S1.

Stable carbon ($\delta^{13}$C), nitrogen ($\delta^{15}$N), and sulphur ($\delta^{34}$S) isotope analyses were performed on extracted collagen and freeze-dried tissue samples at the Scottish Universities Environmental Research Centre (SUERC) and the Royal Netherlands Institute for Sea Research (NIOZ), respectively, using continuous-flow isotope ratio mass spectrometers. Isotopic compositions were normalised using IAEA and in-house standards. Full methodological details and standards used are provided in Appendix S1.

### Data analysis

ZooMS biomarker identification using MALDI-TOF and LC–MS/MS followed the data analysis steps outlined in [15], additional details can be found in Appendix S1. Stable isotope results were quality-controlled using previously published criteria for archaeological collagen [30,31]. To compare modern and ancient specimens, modern $\delta^{13}$C values were adjusted for the Suess effect, the depletion of atmospheric $\delta^{13}$C due to fossil fuel emissions, using Dombrosky's [32] model, applying a correction factor of +2 units. The stable isotope data were visualised with scatterplots, and linear relationships between trophic position ($\delta^{15}$N), length and weight were analysed. ANOVA and post hoc testing were carried out in R [33] to test for differences in $\delta^{13}$C, $\delta^{15}$N and $\delta^{34}$S between Dalaman, Samandağ and Tell el-Burak, we didn't include Tell Fadous in statistical testing as there is just one data point. All visualisations were conducted in R using ggplot2 [34].

## Results

### Physical characteristics of Dalaman turtles

The sizes and weights of sampled *T. triunguis* specimens are summarised in S1 Table. Most sexed individuals were male ($n = 5$), with only two females. Females were notably lighter than males, who also had larger curved carapace lengths (CCL) and widths (CCW), suggesting possible sexual dimorphism in the population.

### ZooMS

The four reliable ZooMS biomarkers identifying *T. triunguis* are shown in Fig 2. These biomarkers were detected in the MALDI-TOF spectra of all specimens, including those also morphologically identified as *T. triunguis*. One of these

**A.**

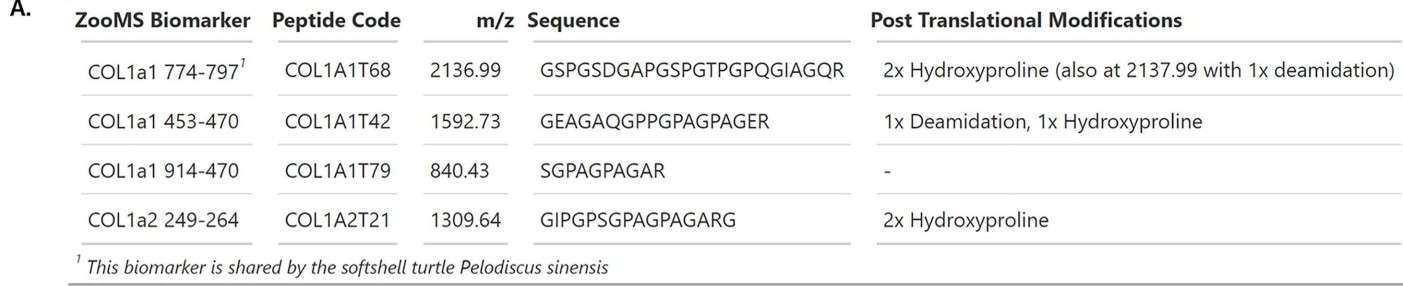

| ZooMS Biomarker | Peptide Code | m/z | Sequence | Post Translational Modifications |
|---|---|---|---|---|
| COL1a1 774-797[1] | COL1A1T68 | 2136.99 | GSPGSDGAPGSPGTPGPQGIAGQR | 2x Hydroxyproline (also at 2137.99 with 1x deamidation) |
| COL1a1 453-470 | COL1A1T42 | 1592.73 | GEAGAQGPPGPAGPAGER | 1x Deamidation, 1x Hydroxyproline |
| COL1a1 914-470 | COL1A1T79 | 840.43 | SGPAGPAGAR | - |
| COL1a2 249-264 | COL1A2T21 | 1309.64 | GIPGPSGPAGPAGARG | 2x Hydroxyproline |

[1] This biomarker is shared by the softshell turtle Pelodiscus sinensis

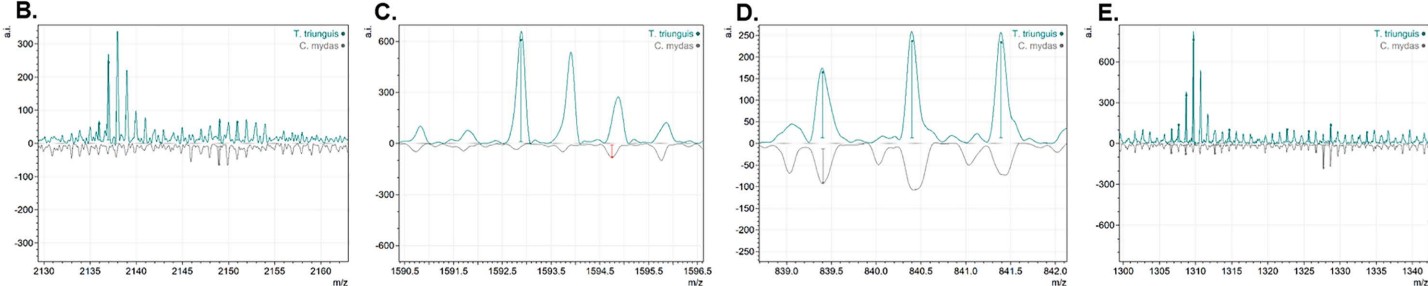

**Fig 2. *T. triunguis* biomarkers.** (**A**) ZooMS biomarkers identified in this study that differentiate Trionyx triunguis from all sea turtle species, as well as seven additional turtle species with COL1 sequences available on NCBI: Chelonoidis abingdonii, Chrysemys picta bellii, Gopherus evgoodei, Mauremys reevesii, Pelodiscus sinensis, Terrapene carolina triunguis, and Trachemys scripta elegans. (**B-E**) Spectra of the four identified biomarkers for T. triunguis (upper), compared to spectra of the sea turtle Chelonia mydas (lower) for reference.

biomarkers, COL1α1 774–797, is shared with the softshell turtle *Pelodiscus sinensis*. However, given that these two species do not have overlapping ranges, this shared biomarker does not pose an issue in this context. Additional ZooMS biomarkers that were either not unique to *T. triunguis* or less reliable are displayed in S2 Fig. Peptide sequences were verified by LC-MS/MS on a museum reference specimen (S2 Fig).

## Stable isotope analyses

Stable isotope values for all specimens are presented in S2 Table. All specimens fell within the C/N quality control range of 2.9–3.6 [30,31]. However, seven of the fifteen specimens failed at least one sulphur quality control criterion (%S, C:S, or N:S) established for archaeological mammals (%S of 0.28±0.07%, C of 600±300, N of 200±100; [31]), used because the equivalent for reptiles is not available. Specifically, three ancient and four modern specimens did not meet these criteria. Pearson's correlations (S3 Table) showed no relationship between sulphur isotopic composition ($\delta^{34}S$) and QC indicators (molar C:S, molar N:S, and %S), which suggested no post-mortem alteration of the collagen [35].

The $\delta^{13}C$ values obtained from the epidermis in modern *T. triunguis* specimens had a narrow range from −22.3‰ to −20.3‰, except for one individual from Samandağ, which had a $\delta^{13}C$ value of −12.7‰. In contrast, the range of $\delta^{13}C$ values among the ancient specimens was much wider (−24.3‰ to −12.4‰, Fig 3A, S2 Table). The three individuals from the ancient site at Tell el-Burak displayed highly variable $\delta^{13}C$ values. However, the sample size is too small to draw strong conclusions. ANOVA testing did not show any statistically significant differences in $\delta^{13}C$ between sites (S4 Table).

The $\delta^{15}N$ values in modern specimens varied by geographic location (Fig 3A). Specimens from Samandağ had higher $\delta^{15}N$ values, ranging from 12.2‰ to 15.3‰, compared to those from Dalaman, which ranged from 5.9‰ to 8.8‰. In the ancient specimens from Tell el-Burak, one individual had a much lower $\delta^{15}N$ value of 3.5‰, while the other two had values of 9.3‰ and 10.6‰. In modern specimens from Dalaman (n=6), $\delta^{15}N$ values were strongly positively correlated with

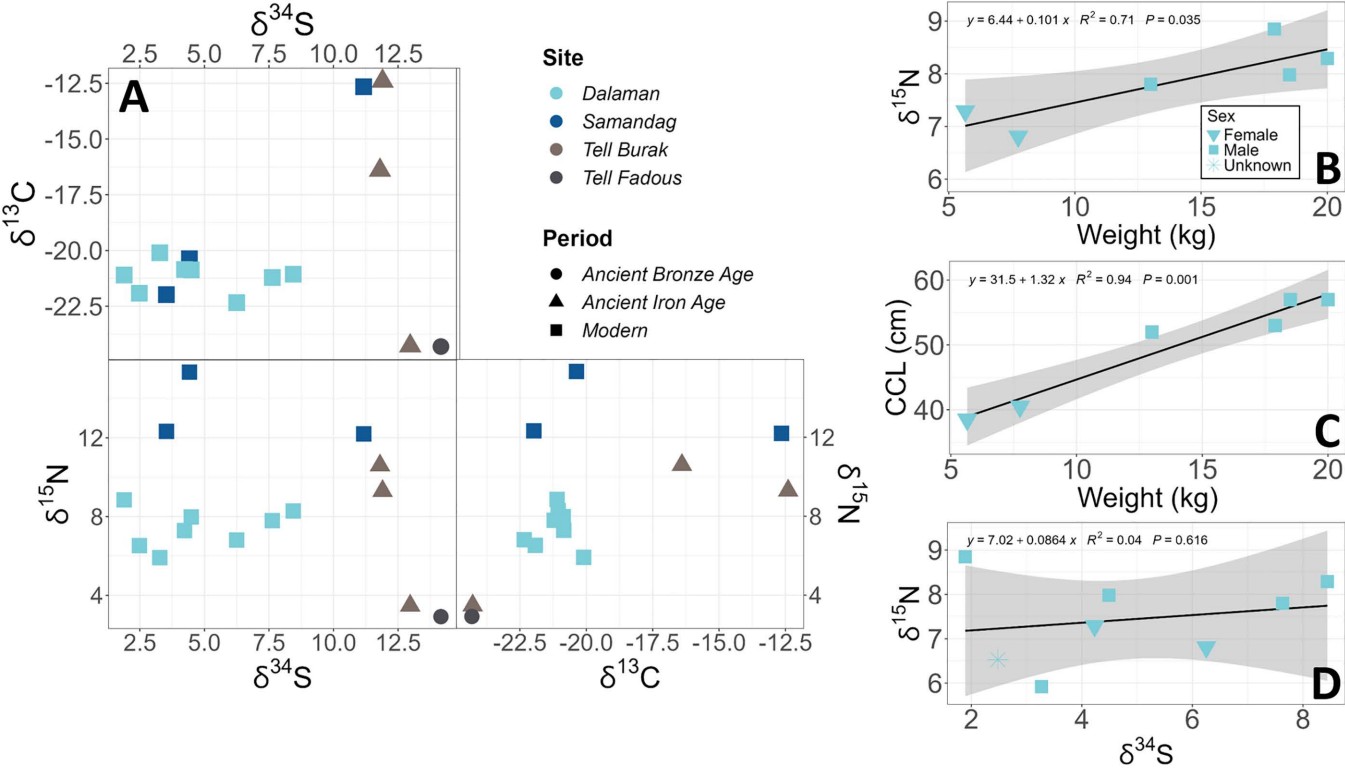

**Fig 3. Stable isotope ecology. (A)** Scatterplot of stable isotope values ($\delta^{13}C$, $\delta^{15}N$, $\delta^{34}S$) of ancient and modern T. triunguis samples. **(B)** Linear regression of weight (kg) of modern Dalaman *T. triunguis* turtles vs $\delta^{15}N$ of skin epidermis. **(C)** Linear regression of weight (kg) vs curved carapace length (cm) in the same individuals. **(D)** Linear regression of $\delta^{34}S$ vs $\delta^{15}N$ of skin epidermis.

weight ($R^2=0.71$, $p<0.05$; Fig 3B), and weight showed a very strong positive relationship with curved carapace length ($R^2=0.94$, $p<0.001$; Fig 3C). ANOVA and subsequent posthoc testing showed significant differences between $\delta^{15}N$ at Samandağ and Dalaman ($p<0.001$) and Samandağ and Tell el-Burak ($p=0.01$).

$\delta^{34}S$ values were generally higher in ancient specimens compared to modern turtles (Fig 3A), though one modern turtle from Samandağ had $\delta^{34}S$ values similar to those of ancient specimens. Modern specimens had $\delta^{34}S$ values ranging from 2.5‰ to 11.2‰ (S2 Table), while ancient samples ranged from 11.8‰ to 14.2‰. ANOVA and subsequent posthoc testing showed significant differences between $\delta^{34}S$ at Tell el-Burak and Samandağ ($p<0.05$) and Tell el-Burak and Dalaman ($p<0.01$).

## Discussion

Our examination of the modern *T. triunguis* population near Dalaman revealed a predominance of males over females. Although the sample size may be insufficient for definitive conclusions, our findings align with [36], who also reported a male bias in this region. A male-biased sex ratio has also been observed in this species on the west coast of Africa [37]. Softshell turtles have genetic sex determination, therefore we would not expect the sex distribution to be skewed from hatching. Instead it is possible that the observed sex distribution could have been influenced by factors such as thermal sampling bias, niche partitioning or the capture technique [38]. Notably, the two females sampled were significantly smaller and lighter than the males (S1 Table). We observed a positive relationship between turtle weight and $\delta^{15}N$ values (Fig 3C), suggesting that males are feeding at a trophic level half to one level higher than females. Semi-aquatic turtle

species often display male-biased sexual size [39], and it is possible that smaller females are less competitive at accessing higher trophic-level prey, such as carcasses.

Our study also identified ZooMS biomarkers with potential for future research in archaeology, museums or ecology. These biomarkers may be particularly valuable for archaeologists working with faunal assemblages from the Levant, where sea turtles and African softshell turtles are frequently found together and can be difficult to distinguish morphologically when remains are fragmented. Visualising stable isotope data (Fig 3A, S2 Table), we noted that seven specimens failed one or more quality control criteria in the sulphur stable isotope analysis [31]. These criteria are typically used to identify post-mortem alterations in collagen, yet four of the failed specimens were modern and therefore had not undergone such degradation. This discrepancy suggests a need to revisit the application of these criteria for reptiles [15]. Using the methodology from [35], correlations between isotopic compositions and quality control criteria (S3 Table) revealed no significant correlations, further suggesting that the observed deviations are not due to post-mortem alteration. These results suggest that quality control criteria established for mammalian bone collagen may not be wholly applicable to reptilian collagen, and highlight the need for developing reptile-specific criteria to assess reliability of sulphur isotopes from archaeological remains. Potential lipid contamination in the modern skin samples warrant consideration, as lipids are depleted in $^{13}C$ relative to collagen and can thus lower $\delta^{13}C$ values while raising C:N ratios. Guiry and Szpak [40] propose a modern-specific C:N atomic ratio threshold of 3.0–3.38, beyond which $\delta^{13}C$ values may be compromised by residual lipids. Only one of our modern samples falls within this stricter range (S2 Table), however, exploration of any correlation between $\delta^{13}C$ and C:N atomic ratios (S1 Fig) shows no statistically significant relationship, suggesting that lipid contamination is not the principal driver of $\delta^{13}C$ variation in our dataset. Ethanol storage prior to analysis may have facilitated some lipid removal, as ethanol is a mild solvent capable of extracting certain polar lipids over extended periods. Any lipid residues present, would likely have a minial effect; Guiry and Szpak [40] estimate that a 0.5 increase in C:N would result in a $\delta^{13}C$ shift of approximately −1‰, a margin insufficient to alter the broader interpretations of dietary ecology presented here. Additionally, when comparing bone and skin collagen isotope values, Doherty et al. [41] report consistent isotopic offsets in modern terrestrial mammals, with skin being depleted in $^{13}C$ by ~0.7‰ and enriched in $^{15}N$ by ~1.0‰. These offsets, while notable, do not affect our interpretations regarding trophic level or habitat use. Overall, while acknowledging the limitations of sample preservation and tissue type, the isotope data are suitable for cautiously comparing foraging patterns in ancient and modern *T. triunguis*.

The stable isotope results suggest shifts in *T. triunguis* foraging ecology, with modern individuals relying more on anthropogenically influenced diets. While ancient turtles showed a mix of marine and terrestrial signals, modern turtles reflect $C_3$-based (associated with most pasture and traditional agricultural crops, such as wheat and rice) and human-provided resources, whereas Samandağ individuals show higher trophic-level feeding (Fig 3). Ancient *T. triunguis* remains are extremely rare, making this dataset uniquely valuable despite its small size. Similarly, living *T. triunguis* are both rare and challenging to study [6], as their large size and aggressive nature make them difficult to capture and handle. However, the severe limitations in sample size and site representation necessitate caution in interpreting these differences. While statistical comparisons are limited, the observed data provide insights into past and present foraging, and susceptibility to human impact.

The narrow $\delta^{13}C$ range among the modern *T. triunguis* specimens suggest a diet primarily influenced by $C_3$ plants, which include most temperate-climate crops such as wheat, barley and most pasture in temperate areas. $C_3$ plants follow a photosynthetic pathway that results in lower $\delta^{13}C$ values compared to $C_4$ plants, such as maize or sugarcane, which have a more enriched $\delta^{13}C$ signature [42]. The $\delta^{13}C$ values from Dalaman *T. triunguis* are the same as modern sheep from the Aegean coast of Turkey [43], sheep carcasses are the predominant carrion expected in the waterways of this region. The findings indicate human-modified environments likely contributed to the dietary signatures observed in these turtles, warranting further investigation into the extent of anthropogenic influence on their feeding ecology. A single exception was the specimen from Samandağ (specimen Samandağ_3), which had higher $\delta^{13}C$ values in line with marine foraging. Two

of the ancient specimens, one from Tell-Fadous-Kfarabida (W68) and one from Tell el-Burak (W44) have depleted $\delta^{13}C$ values which could have been due to agricultural influence in *T. triunguis* diets. Tell-Fadous-Kfarabida's barley (*Hordeum vulgare*) was particularly depleted in $\delta^{13}C$, likely due to higher rates of precipitation at the time (Hermann [44]). At Tell el-Burak, grape (*Vitis vinifera*) remains, comprising 35.5% of the assemblage [45], showed $\delta^{13}C$ depletion compared to grains [46], indicating a potentially substantial agricultural influence on *T. triunguis'* diet at the time. In contrast, the other two Tell el-Burak individuals (W45 and W49) resembled the modern turtle Samandağ_3, with $\delta^{13}C$ values suggesting marine foraging.

The low $\delta^{15}N$ in ancient Tell el-Burak samples W44 and W68 complement the theory that these ancient turtles were influenced by agriculture. At the same site, a contemporary freshwater detritus-feeding vertebrate (*Clarias gariepinus*) has higher $\delta^{15}N$ (9.54) values [26]. Archaeobotanical remains from nearby Tel Kabri, like olive (*Olea europaea*) and grapes have negative $\delta^{15}N$ values (Nicoli' et al. 2023), supporting the idea of an agriculture influenced diet in these turtles. Conversely, W45 and W49 showed higher $\delta^{15}N$ values, indicative of a wild diet possibly influenced by marine environments. Modern $\delta^{15}N$ values varied; Dalaman specimens showed similar $\delta^{15}N$ values, whereas Samandağ turtles fed at higher trophic levels or in a food web with a higher $\delta^{15}N$ baseline (Fig 3A). Generally, food chains are longer in aquatic environments, leading to higher $\delta^{15}N$ values in omnivores and carnivores who have wild aquatic diets [47]. It is possible that the Dalaman population is primarily anthro-dependent, feeding on human-discarded carrion, or fed by tourists, while Samandağ turtles consume higher-level aquatic prey. Taken together, these results suggest that foraging patterns may differ between populations and between time periods, although the small number of samples limits robust comparison.

Sulphur stable isotope ($\delta^{34}S$) results showed significant variation between Tell el-Burak and the modern sites. Ancient Levant specimens had higher $\delta^{34}S$ values than modern Turkish specimens, except for Samandağ_3, which matched the ancient specimens (Fig 3A). Elevated $\delta^{34}S$ levels indicate marine influence [48]. For the modern specimen Samandağ_3, this is consistent with the $\delta^{13}C$ and $\delta^{15}N$ results, which also suggest marine foraging. The $\delta^{34}S$ interpretation for ancient samples is complex due to the effect of sea spray, potentially affecting $\delta^{34}S$ up to 30 km inland [48]. Modern specimens, except Samandağ_3, did not exceed $\delta^{34}S$ values seen in ancient and modern freshwater fish from outside the sea spray range [49]. Therefore, despite Dalaman turtles being caught within two kilometres of the sea, minimal sea spray effect is suggested. Although all ancient specimens exhibited relatively high and similar $\delta^{34}S$ values, only W45 and W49 also displayed elevated $\delta^{13}C$ and $\delta^{15}N$ values indicative of marine feeding behaviour. The other two ancient specimens exhibited freshwater isotope values; however, the absence of freshwater fish, bivalves, or gastropod remains at Tell Fadous-Kfarabida suggests that freshwater resources were not available locally. This raises the possibility that the two distinct specimens, W44 and W68, were imported from a strongly freshwater environment, such as an inland lake or the Nile River. Modern Dalaman specimens showed varied $\delta^{34}S$ values, however the lack of significant relationship to $\delta^{15}N$ indicates factors beyond size or sex influence $\delta^{34}S$ values. Samandağ turtles with low $\delta^{34}S$ values likely spent most of their lives in freshwater.

Our study suggests that *T. triunguis* diets may vary among populations and between past and present contexts, with some evidence for marine foraging and anthropogenic influence. Modern turtles from Samandağ exhibit significantly higher $\delta^{15}N$ values than those from Dalaman, suggesting a shift toward higher trophic-level feeding. In contrast, ancient Tell el-Burak turtles show no significant $\delta^{15}N$ difference from Dalaman but exhibit significantly elevated $\delta^{34}S$ values, indicating stronger marine influences. These patterns suggest that while prehistoric turtles had varied diets, including marine foraging, modern populations are more dependent on human-altered resources. The varied *T. triunguis* foraging highlights their adaptability but also their vulnerability to human-driven ecological changes. Anthropogenic influences on *T. triunguis* diets may pose risks through exposure to agricultural pollutants and toxins [7]. Given the small sample sizes, these patterns are preliminary, but they provide useful insights for future research and management of this little-studied species.

## Supporting information

**S1 File. Additional descriptions, methods and results.**
(DOCX)

**S1 Table. Curved carapace length (CCL), curved carapace width (CCW), weight (kg), and sex of the eight Trionyx triunguis sampled in Dalaman.**
(PNG)

**S2 Table. Sample information and stable isotope measurements of modern and ancient *Trionyx triunguis* specimens.** Quality control criteria which fell outside the range proposed for archaeological collagen [30,31] are displayed in red.
(PNG)

**S3 Table. Results of Pearson's product-moment correlation tests performed to evaluate the relationships between $\delta^{34}S$ and three quality control (QC) criteria: the molar carbon to sulphur ratio (CS Molar), the molar nitrogen to sulphur ratio (NS Molar), and the percentage sulphur (%S).** Provided are the t-statistic, degrees of freedom, p-value indicating significance, 95% confidence interval for the correlation coefficient, and the Pearson correlation coefficient.
(PNG)

**S4 Table. Post hoc testing of pairwise comparisons of $\delta^{13}C$, $\delta^{15}N$, and $\delta^{34}S$ isotope values between sites (Samandağ, Dalaman, and Tell Burak).** Differences between site pairs are reported alongside the 95% confidence intervals (CI) and associated p-values. Significant differences ($p < 0.05$) are in bold.
(PNG)

**S1 Fig. Pearson's product-moment correlation tests of $\delta^{13}C$ vs C:N ratio in A: all modern T. triunguis skin samples, and B: all modern samples excluding Samandağ_3 which appears to have a separate ecological niche.**
(PNG)

**S2 Fig. A) Summary of ZooMS biomarkers identified in this study.** Biomarkers are categorised as either shared with sea turtles (Harvey et al. 2019), shared with Pelodiscus sinensis (a softshell turtle native to China and Taiwan), or unique to T. triunguis. *In this context, "unique" refers to biomarkers that were not found in any of the tested turtle COL1 sequences. (B-E) LC-MS/MS spectra of the four reliable biomarkers presented in Figure 2 of the manuscript, visualised using pBuild. The spectra show coverage of Y and B ions, along with the amino acid substitutions and any post-translational modifications.
(PNG)

## Acknowledgments

We gratefully acknowledge Dr. Simona Ceriani for her valuable advice on skin biopsy processing, and M. Munoz-Alegre for her mass-spectrometry support. Our sincere thanks go to Dr. Bektaş Sönmez for providing the Samandağ specimens. Skin biopsies were exported to the University of Groningen under CITES permit 22NL305802/11. Archaeological samples are stored in the zooarchaeological collection of the Groningen Institute of Archaeology (University of Groningen). Our research would not have been possible without the work of the Tell Fadous-Kfarabida and Tell el-Burak excavation teams. Many thanks to S. Ikram. We also appreciate D. Klingberg Johansson and the National History Museum of Denmark for permitting us to sample *T. triunguis* for proteomic analysis.

## Author contributions

**Conceptualization:** Willemien de Kock, Canan Çakırlar, Oguz Turkozan.

**Formal analysis:** Willemien de Kock.

**Investigation:** Marcel T.J. van der Meer, Ronald van Bommel, Alberto J. Taurozzi, Matthew Von Tersch, Morten E. Allentoft, Meaghan Mackie, Max Ramsøe, Matthew Collins, Michelle Alexander, Oguz Turkozan.

**Methodology:** Willemien de Kock, Per J. Palsbøll.

**Supervision:** Morten E. Allentoft, Michelle Alexander, Per J. Palsbøll, Canan Çakırlar, Oguz Turkozan.

**Writing – original draft:** Willemien de Kock.

**Writing – review & editing:** Willemien de Kock, Marcel T.J. van der Meer, Ronald van Bommel, Alberto J. Taurozzi, Matthew Von Tersch, Morten E. Allentoft, Meaghan Mackie, Max Ramsøe, Matthew Collins, Michelle Alexander, Per J. Palsbøll, Canan Çakırlar, Oguz Turkozan.

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
