## [Decision Letter · Decision Letter 0]

24 Sep 2025

Dear Dr. de Kock,

Thank you for submitting your manuscript to PLOS ONE. After careful consideration, we feel that it has merit but does not fully meet PLOS ONE’s publication criteria as it currently stands. Therefore, we invite you to submit a revised version of the manuscript that addresses the points raised during the review process.

We look forward to receiving your revised manuscript.

Kind regards,

Vitor Hugo Rodrigues Paiva, Ph.D.

Academic Editor

PLOS ONE

Journal Requirements:

[The first author received support from the Marie Skłodowska-Curie Innovative Training Network SeaChanges, funded by the European Union’s Horizon 2020 research and innovation programme (Marie Skłodowska-Curie grant agreement No. 813383), and from the Netherlands Sectorplan Social Sciences and Humanities (SSH, 2022). This study was funded by the Royal Netherlands Academy of Arts and Sciences (KNAW) Ecology Fund under grant number KNAWWF/747/ECO2021-17.].

[We gratefully acknowledge Dr. Simona Ceriani for her valuable advice on skin biopsy processing, and M. Munoz-Alegre for her mass-spectrometry support. Our sincere thanks go to Dr. Bektaş Sönmez for providing the Samandağ specimens. Skin biopsies were exported to the University of Groningen under CITES permit 22NL305802/11. Archaeological samples are stored in the zooarchaeological collection of the Groningen Institute of Archaeology (University of Groningen). Our research would not have been possible without the work of the Tell Fadous-Kfarabida, and Tell el-Burak excavation teams. Many thanks to S. Ikram. We also appreciate D. Klingberg Johansson and the National History Museum of Denmark for permitting us to sample T. triunguis for proteomic analysis. The first author received support from the Marie Skłodowska-Curie Innovative Training Network SeaChanges, funded by the European Union’s Horizon 2020 research and innovation programme (Marie Skłodowska-Curie grant agreement No. 813383), and from the Netherlands Sectorplan Social Sciences and Humanities (SSH, 2022). This study was funded by the Royal Netherlands Academy of Arts and Sciences (KNAW) Ecology Fund under grant number KNAWWF/747/ECO2021-17.]

[The first author received support from the Marie Skłodowska-Curie Innovative Training Network SeaChanges, funded by the European Union’s Horizon 2020 research and innovation programme (Marie Skłodowska-Curie grant agreement No. 813383), and from the Netherlands Sectorplan Social Sciences and Humanities (SSH, 2022). This study was funded by the Royal Netherlands Academy of Arts and Sciences (KNAW) Ecology Fund under grant number KNAWWF/747/ECO2021-17.]

7. We note that Figure 1 in your submission contains map/satellite images which may be copyrighted. All PLOS content is published under the Creative Commons Attribution License (CC BY 4.0), which means that the manuscript, images, and Supporting Information files will be freely available online, and any third party is permitted to access, download, copy, distribute, and use these materials in any way, even commercially, with proper attribution. For these reasons, we cannot publish previously copyrighted maps or satellite images created using proprietary data, such as Google software (Google Maps, Street View, and Earth). For more information, see our copyright guidelines: http://journals.plos.org/plosone/s/licenses-and-copyright.

8. Please include captions for your Supporting Information files at the end of your manuscript, and update any in-text citations to match accordingly. Please see our Supporting Information guidelines for more information: http://journals.plos.org/plosone/s/supporting-information .

Reviewers' comments:

Reviewer's Responses to Questions

**Comments to the Author**

1. Is the manuscript technically sound, and do the data support the conclusions?

Reviewer #1: Yes

Reviewer #2: No

2. Has the statistical analysis been performed appropriately and rigorously?

Reviewer #1: Yes

Reviewer #2: Yes

3. Have the authors made all data underlying the findings in their manuscript fully available?

Reviewer #1: Yes

Reviewer #2: Yes

4. Is the manuscript presented in an intelligible fashion and written in standard English?

Reviewer #1: Yes

Reviewer #2: Yes

Reviewer #1: This manuscript presents an isotope study of modern and ancient (Middle Holocene) softshell turtle remains in the Eastern Mediterranean. The ancient remains were taxonomically confirmed by ZooMS, consisting one of the first such applications for reptiles. Then, C, N, and S isotopes allowed the authors to characterize turtle diets and compare the recent specimens to the archaeological ones. The differences were interpreted as greater human impact on turtle food availability in the modern samples, relative to the archaeological “baseline”. I liked the study concept and design. The manuscript is interesting, original, and well-written, and I support its publication. I have a few recommendations, detailed below, which can be easily dealt with.

A point that should be emphasized is that the study is based on a small number of modern samples, and an even smaller number of archaeological ones (n=4, from two different sites and two different periods that are centuries apart). In a way, the greater diversity of the archaeological specimens can be parsimoniously explained by this (inevitable) way of sampling. While the authors justify this convincingly in the text, the small samples – that moreover are very scattered temporally and spatially – are a major limitation of this study. In my opinion, the readers should be aware of this from the beginning and therefore I recommend to mention the N in the abstract, e.g., “Ancient turtles (n=4) exhibit more diverse diets…” and mention the justification for the small samples earlier on. This does not detract from the importance of the study, but is necessary to properly appreciate its validity.

The archaeological samples actually come from the Lebanese coast, not from Turkey (as the modern samples). This should also be mentioned in the abstract. The study contributes interesting information on turtles deposited in Bronze/Iron Age sites in Lebanon, so Zooarchaeologists looking for such information should be able to find it.

Line 105: the phrasing is confusing, as the authors portray a specimen from the 1950s as “the oldest known Holocene specimen”, but then mention much older (archaeological) ones.

Line 139: the reference should be Fig. 1, not Fig. 2.

Methods: I find the bold font unnecessary and distracting.

Reviewer #2: In general, this is a well-structured and interesting paper. For the first time, the authors use stable isotope and ZooMS analysis to explore dietary shifts in modern and ancient African softshell turtles. The novelty of this research is clear, and the new ZooMS markers are valuable additions to the field. The authors have also tried to make their data as accessible as possible, while clarifying several limitations of their research.

My main concern regarding this research is that the limited data presented in the paper cannot answer the three questions raised in the introduction.

• Whether their foraging behaviours favour marine or freshwater environments?

With only eight modern samples and four ancient samples, it is impossible to answer this question.

• How these behaviours may have changed over time?

Again, with only eight modern samples, three Iron Age samples and one Bronze Age sample, it is impossible to answer this question. It makes no sense to discuss temporal changes.

• If this behaviour is universal or specific to certain populations

I did not see the authors discuss this point at all.

Therefore, I ask the authors to add more ancient samples, although I don’t know if any are available. With the current limited dataset, the pattern is unclear to me. I would not recommend accepting this paper since the argument cannot be supported by solid evidence. However, under the current framework, it would be a high-quality paper if more data were added.

Other comments

1. In the introduction, explain why the authors specifically chose the C, N, and S isotopes, and how and why they work. This is particularly important for the S isotope, since it is not commonly used in archaeological research.

2. Introduce the ecological and environmental context of the sites under study.

3. Are there any differences between Tell el-Burak and Tell Burak?

4. Differentiate between the sexes in Figures 3B–D.

5. Figure 2 is difficult to read. Please try to improve it.

**Do you want your identity to be public for this peer review?** For information about this choice, including consent withdrawal, please see our Privacy Policy

Reviewer #1: No

Reviewer #2: No

---

## [Decision Letter · Decision Letter 1]

9 Dec 2025

Stable Isotope Evidence of Anthropocene Disruption in African Softshell Turtle Foraging

PONE-D-25-32660R1

Dear Dr. de Kock,

We’re pleased to inform you that your manuscript has been judged scientifically suitable for publication and will be formally accepted for publication once it meets all outstanding technical requirements.

Kind regards,

Vitor Hugo Rodrigues Paiva, Ph.D.

Academic Editor

PLOS One

Additional Editor Comments (optional):

Reviewers' comments:

Reviewer's Responses to Questions

**Comments to the Author**

Reviewer #1: All comments have been addressed

Reviewer #2: All comments have been addressed

2. Is the manuscript technically sound, and do the data support the conclusions?

Reviewer #1: Yes

Reviewer #2: Partly

3. Has the statistical analysis been performed appropriately and rigorously?

Reviewer #1: Yes

Reviewer #2: N/A

4. Have the authors made all data underlying the findings in their manuscript fully available?

Reviewer #1: Yes

Reviewer #2: Yes

5. Is the manuscript presented in an intelligible fashion and written in standard English?

Reviewer #1: Yes

Reviewer #2: Yes

Reviewer #1: The authors satisfactorily revised the manuscript. Please refer to their letter for details.

Reviewer #2: The authors have tried their best to address the issues I mentioned before, I don't have further comments.

**Do you want your identity to be public for this peer review?** For information about this choice, including consent withdrawal, please see our Privacy Policy

Reviewer #1: No

Reviewer #2: **Yes:** Li Tang

---

## [Editor Report · Acceptance letter]

PONE-D-25-32660R1

PLOS One

Dear Dr. de Kock,

I'm pleased to inform you that your manuscript has been deemed suitable for publication in PLOS One. Congratulations! Your manuscript is now being handed over to our production team.

Kind regards,

on behalf of

Dr. Vitor Hugo Rodrigues Paiva

Academic Editor

PLOS One